# Efficacy of Subsequent Therapy in Patients with Hormone-Positive Advanced Breast Cancer with Disease Progression After CDK4/6 Inhibitor Therapy: Multicenter Real-Life Data

**DOI:** 10.3390/jcm14207376

**Published:** 2025-10-18

**Authors:** Buket Şahin Çelik, Aslı Geçgel, Oğuzcan Özkan, Nargiz Majidova, Buket Erkan Özmarasalı, Gözde Ağdaş, İsmail Bayrakçı, Türkkan Evrensel, Erhan Gökmen

**Affiliations:** 1Ege University Faculty of Medicine Hospital, 35100 İzmir, Türkiye; dr.asltrgt@gmail.com (A.G.);; 2Marmara University Faculty of Medicine Hospital, 34854 İstanbul, Türkiye; 3Uludağ University Faculty of Medicine Hospital, 16059 Bursa, Türkiye; 4Eskişehir Osmangazi University Faculty of Medicine Hospital, 26040 Eskişehir, Türkiye; 5Trakya University Faculty of Medicine Hospital, 22030 Edirne, Türkiye

**Keywords:** metastatic breast cancer, CDK4/6 inhibitor, everolimus, endocrine therapy, chemotherapy, survival outcomes

## Abstract

**Objectives**: The aim of this study was to evaluate the effect of different systemic therapies after CDK4/6 inhibitor therapy on survival outcomes in HR+/HER2− metastatic breast cancer (MBC) patients. **Methods**: In this retrospective multicenter study, patients who continued chemotherapy (CT), everolimus + endocrine therapy (HT), and other hormonotherapy after treatment with a CDK4/6 inhibitor were compared. Clinicopathological data and survival outcomes were analyzed. Statistical analyses were performed using the SPSS v25 program, survival analyses were performed using the Kaplan–Meier method, and comparisons were made using the log-rank test. **Results**: A total of 145 patients were included in the study. The groups were similar in terms of baseline characteristics such as age, menopausal status, histology, stage, adjuvant treatment status, and metastatic spread pattern. The rate of recurrent disease was significantly higher in the CT and Everolimus + HT groups compared to the “Other” group (*p* = 0.027). However, there was no significant difference between the groups in terms of PFS and OS in the general population (*p* > 0.05). In subgroup analyses, OS was significantly longer in the everolimus + HT group compared to the CT group in those with recurrence duration ≥ 1 year and stable disease course > 6 months during CDK4/6 inhibitor treatment (*p* = 0.010 and *p* = 0.039). **Conclusions**: Although there was no significant difference in overall survival regarding the choice of treatment after a CDK4/6 inhibitor, everolimus + endocrine therapy was observed to have a positive effect on survival in some subgroups. This finding supports individualized treatment.

## 1. Introduction

Endocrine-based treatment approaches are the primary recommendation for advanced HR-positive, human epidermal growth factor receptor 2 (Her2)-negative patients with no evidence of visceral crisis [1,2]. Hormone receptor (HR) positive breast cancers account for approximately 70% of all breast cancer cases, and this subgroup is of particular importance in clinical management [3]. However, the progression-free survival (PFS) achieved with monotherapy endocrine therapies in this patient group is limited to a median of 12 months [4]. In recent years, it has been demonstrated that the addition of CDK4/6 inhibitors to endocrine therapy provides a significant and marked improvement in PFS [5]. Because resistance to endocrine therapy plays an important role in treatment failure, activation of the cyclin-dependent kinase 4/6 inhibitor (CDK4/6 inhibitor) pathway is one of the main mechanisms in the development of endocrine resistance [4]. Therefore, administering CDK4/6 inhibitors with endocrine therapy increases the success of endocrine therapy. However, for HR+/HER2− advanced breast cancer patients who progress under CDK4/6 inhibitor and endocrine therapy, there is no standard treatment recommended at the category 1 level in international guidelines, unless there is a positive mutation analysis result. The aim of this study was to determine the appropriate treatment after CDK4/6 inhibitor therapy.

Unless there is a visceral crisis, monotherapy endocrine therapies, such as fulvestrant, exemestane, or combinations with mTOR inhibitors, are preferred. Furthermore, the combination of alpelisib + fulvestrant is recommended as a suitable option in patients with PIK3CA mutation. In recent years, multicenter, real-life studies have shown that endocrine-based treatments can be just as effective as chemotherapy in the post-CDK4/6 inhibitor period. However, in clinical practice, chemotherapy after CDK4/6 inhibitors may be preferred in some patients even in the absence of visceral crisis. Therefore, data and real-life experiences are needed on subsequent treatment options, especially in patients without mutations and visceral crisis.

In a study published in BMC Cancer involving 53 centers, it was shown that endocrine therapy after CDK4/6 inhibitor treatment can be as effective as chemotherapy [6]. In this multicenter retrospective study, it was observed that, in HR + HER2− advanced breast cancer patients progressing under CDK 4/6 inhibitor therapy, a short duration of CDK4/6 inhibitor treatment increased physicians’ preference for chemotherapy as a subsequent treatment. There was no difference in PFS between the subsequent CT and ET arms. Among patients receiving the CDK4/6 inhibitor as first-line treatment, when endocrine-based therapies were compared with monotherapy and everolimus-based therapies, everolimus-based therapies were associated with longer PFS. Retrospective analyses of TREND, a phase 2 study, showed no difference in duration of treatment between subsequent CT and ET (4.6 vs. 3.7 months), regardless of palbosiclib use [7]. Furthermore, in a study by Xi et al., the overall mean time to treatment failure was as follows: everolimus/exemestane, 13.2 months; single-agent hormone therapy, 3.1 months; and single-agent chemotherapy, 4.1 months. Therefore, ET-based approaches are appropriate options after CDK4/6 inhibitor progression, unless there are cases of rapid disease progression [8].

In patients carrying germline or somatic BRCA mutations, the PARP inhibitors olaparib or talazoparib can be administered and are recommended in the new NCCN guidelines for second-line or more advanced treatments, especially in patients who progress after CDK4/6 inhibitors [9].

In studies such as OLYMPIAD and EMBRACA, these drugs have been shown to provide better progression-free survival (PFS) by about three months compared to chemotherapy [9] Furthermore, in the TBCRC048 study, a 58% response rate to PARP inhibitors was detected in patients with non-BRCA HRR (such as PALB2, ATM, and CHEK2) mutations who were treated after CDK4/6 inhibitor [10].

In patients with PIK3CA or AKT1 activation mutation or PTEN alteration, alpelisib + fulvestrant and capivasertib + fulvestrant combinations are recommended in cases of progression after CDK4/6 inhibitor therapy (second or subsequent steps) [11].

In patients with genetic alterations such as PIK3CA, AKT1, or PTEN mutations, the combination of capivasertib + fulvestrant significantly improved progression-free survival in the CAPItello-291 phase III trial, and this combination was upgraded to a category 1 recommendation in the post- CDK4/6 inhibitor period [12,13]. In HER2 IHC 1+ or 2+/ISH negative (HER2-low) patients, fam-trastuzumab deruxtecan-nxki is recommended as a category 1 and preferred subsequent treatment option [14]. In addition, next-generation oral SERDs (e.g., giredestrant) are emerging in endocrine-resistant patients with ESR1 mutation [15].

To provide insight into treatment options considered after palbosiciclib treatment in Japan, PALOMA-3 evaluated treatment patterns after palbosiclib treatment in Japanese patients participating in clinical trials [16]. PALOMA 3, a randomized clinical trial comparing fulvestrant with fulvestrant + palbosiciclib in previously treated advanced breast cancer patients, showed no difference in treatment duration between BT and ET use after palbosiclib (5.6 vs. 4.3 months). However, it is important to know that none of the patients in the treatment group received everolimus plus exemestane as the first subsequent treatment. Because more real-life data were needed, the other two arms to be compared with chemotherapy in our study were mTOR inhibitor + ET and endocrine therapy only [16]. In short, there is no clear category 1 subsequent treatment recommendation in the NCCN guidelines for patients without mutations and visceral crisis; treatment strategies in this patient subgroup are still controversial [9]. Current NCCN and Japanese Breast Cancer Society (JBCS) guidelines in this subgroup do not include clear and standardized recommendations on the optimal treatment sequence or subsequent treatment options after CDK4/6 inhibitor therapy. Therefore, studies based on real-life data are of great importance in this area.

## 2. Materials and Methods

### 2.1. Study Design and Patient Disposition

This study was a retrospective, multicenter analysis conducted between November 2017 and July 2023 at five different centers located throughout Turkey. The study included female patients diagnosed with HR positive/HER2 negative metastatic breast cancer who underwent systemic therapy after CDK4/6 inhibitor treatment. Inclusion criteria were age 18 years or older, estrogen and/or progesterone receptor levels ≥ 10% in tumor tissue, and at least one systemic therapy (chemotherapy or endocrine-based therapy) after CDK4/6 inhibitor treatment. Patients taking CDK4/6 inhibitors in early-stage disease and patients with HER2 positivity were excluded from the study. Patients with actionable mutations (e.g., *PIK3CA*, *BRCA1/2*) were excluded, as they would be candidates for targeted therapies not comparable to the treatment arms investigated in this study. Ethics committee approval was obtained from the relevant centers for all patients. The analysis of the data was performed using IBM SPSS Statistics, version 25.0 (IBM Corp., Armonk, NY, USA).

### 2.2. Data Acquisition and Clinical Variables

Demographic data (age, menopausal status), clinicopathological characteristics (date of diagnosis and metastasis, tumor subtype, stage, metastasis sites), treatment protocols, treatment response, and survival data were obtained retrospectively from electronic medical record systems and patient files. All data were organized using a standard data form to ensure inter-center alignment before analysis.

### 2.3. Treatment Arms and Subgroup Classifications

The patients were divided into three main groups according to the initial systemic therapy they received after CDK4/6 inhibitor treatment:Chemotherapy (CT) groupEverolimus + endocrine therapy (HT) group (combination with mTOR inhibitor)Other treatments (monotherapy endocrine therapy or different agents)

Additionally, the time of recurrence (<1 year, ≥1 year), time to progression on CDK4/6 inhibitors (≤6 months, >6 months), presence of visceral metastases, and treatment response were evaluated in subgroup analyses.

Recurrence > 1 year refers to patients who relapsed ≥12 months after completion of adjuvant endocrine therapy and are therefore considered *endocrine-sensitive*. Recurrence < 1 year refers to patients who relapsed within the first 12 months after the completion of adjuvant endocrine therapy and represents *endocrine-resistant* patients.

Patients who experienced disease progression during adjuvant endocrine therapy were not included in this study, as this subgroup was not analyzed separately.

### 2.4. Statistical Analysis

The normality of the data distribution was tested using the Shapiro–Wilks test. If the data showed a normal distribution, parametric tests were applied, and mean ± standard deviation values are given as descriptive statistics. In the event that the data did not show a normal distribution, non-parametric tests were applied, and the descriptive statistics are given as the median (min–max) value. Inter-group comparisons were performed via analysis of variance or the Kruskal–Wallis test in the case of more than two groups. Inter-group comparisons of categorical data were performed by chi-square test, Fisher’s exact test, or Fisher–Freeman–Halton test. Categorical data are presented as frequency and percentage. Survival analyses were performed via Kaplan–Meier analysis, and survival times were calculated through a log-rank test. The descriptive statistics of survival times are presented as mean ± standard error. The statistical significance level was defined as α = 0.05. The analysis of the data was performed using the SPSS v25 program.

The authors acknowledge the assistance of ChatGPT (GPT-5, OpenAI, San Francisco, CA, USA) for language editing and formatting support. The authors have reviewed and verified all generated content. ChatGPT was not used in any way for data analysis or result generation.

### 2.5. Ethics Committee Approval

This study was approved by the ethics committee decisions No. 24-1.1T/52 dated 29 January 2024 and No. 24-3T/84 dated 7 March 2024. The research was conducted in accordance with the Declaration of Helsinki.

## 3. Results

There were no statistically significant differences in age, number of metastases, and KI67 levels depending on the treatment administered after CDK4/6 inhibitors (Table 1).

### 3.1. Distribution of Demographic, Pathological, and Treatment Response Data by Post-CDK4/6 Inhibitor Therapy

There were no statistically significant differences in the distribution of menopause, pathologic diagnosis, and recurrence depending on the treatment administered after CDK4/6 inhibitors (Table 2).

A statistically significant difference was found in the distribution of recurrent disease depending on the treatment administered after CDK4/6 inhibitors. In pairwise comparisons made due to significant differences between the groups, a significant difference was found between CT and Everolimus + HT and Other. The rate of recurrent disease was higher in the CT and Everolimus + HT groups, and lowest in the Other group. There was no significant difference between the CT and Everolimus + HT groups (Table 2).

There were no statistically significant differences in stage or T and N distributions depending on the treatment administered after CDK4/6 inhibitors (Table 3).

There were no statistically significant differences in the distribution of primary operated, adjuvant CT, adjuvant HT, ai, progression under adjuvant treatment, metastatic first-line metastatic area, metastatic area at the time of diagnosis, progression while receiving CDK4/6 inhibitors, endocrine with MTOR, response to treatment after CDK4/6 inhibitors, metastatic area developed post-CDK4/6 inhibitor, and CDK4/6 inhibitors depending on the post-CDK4/6 inhibitor treatment (Table 4).

A statistically significant difference was found in the presence of de novo disease compared to post-CDK4/6 inhibitor therapy. A significant difference was found between Everolimus + HT and Other in the pairwise comparisons. The de novo disease prevalence rate was higher in the Other group than in the Everolimus + HT group. No significant difference was found between CT and Everolimus + HT or between CT and Other (Table 4).

A statistically significant difference was found between PR status and post-CDK4/6 inhibitor therapy. In pairwise comparisons made due to significant differences between the groups, a significant difference was found between CT and Everolimus + HT. The everolimus + HT group exhibited a higher positivity rate than the CT group. No significant differences were found between Other and CT and Everolimus + HT (Table 4).

### 3.2. Survival Analysis by Subsequent Treatment in Patients Who Progressed While Receiving CDK4/6 Inhibitors by CDK4/6 Inhibitor Initiation Line

No significant difference was found in PFS duration by treatment administered after CDK4/6 inhibitors (Table 5).

No significant difference was found in OS duration by treatment administered after CDK4/6 inhibitors (Table 5).

No significant difference was found in PFS duration by treatment after CDK4/6 inhibitors as first-line treatment (Table 6).

No significant difference was found in PFS duration by treatment after CDK4/6 inhibitors as second-line treatment. (Table 6).

No significant difference was found in PFS duration by treatment after CDK4/6 inhibitors as third-line treatment (Table 6).

No significant difference was found in OS duration by treatment after CDK4/6 inhibitors as first-line treatment (Table 6).

No significant difference was found in OS duration by treatment after CDK4/6 inhibitors as second-line treatment (Table 6).

No significant difference was found in OS duration by treatment after CDK4/6 inhibitors as third-line treatment (Table 6).

### 3.3. Analysis of Recurrence Time and Progression Time While on CDK4/6 Inhibitors

There was no significant difference in PFS duration in patients with recurrence at <1 year compared to treatment after CDK4/6 inhibitor treatment (Table 7).

There was no significant difference in PFS duration in patients with recurrence at ≥1 year compared to treatment after CDK4/6 inhibitor treatment (Table 7).

A significant difference was found in OS duration in patients with recurrence at ≥1 year compared to treatment after CDK4/6 inhibitor treatment. Everolimus + HT patients had a higher OS duration than CT patients (Table 7) (Figure 1).

No significant difference was found in PFS for progression in the first 6 months or less while receiving CDK4/6 inhibitors compared to treatment after CDK4/6 inhibitors (Table 8).

No significant difference was found in PFS duration for those who developed progression after 6 months while receiving CDK4/6 inhibitors compared to treatment after CDK4/6 inhibitors (Table 8).

No significant difference was found in PFS times for progression in the first 6 months or less while receiving CDK4/6 inhibitors compared to treatment after CDK4/6 inhibitors (Table 8).

A significant difference was found in the OS duration of those who developed progression after 6 months while receiving CDK4/6 inhibitors compared to treatment after CDK4/6 inhibitors. Everolimus + HT patients had a higher OS duration than CT patients (Table 8) (Figure 2).

### 3.4. Subgroup Analysis by Recurrence Time and Progression Time While Taking CDK4/6 Inhibitors, According to CDK4/6 Inhibitor Usage Line

There is no statistically significant difference in PFS in those with recurrence at more than 1 year in first-line patients by treatment after CDK4/6 inhibitors (Table 9).

There is no statistically significant difference in OS in those with recurrence at more than 1 year in first-line patients by treatment after CDK4/6 inhibitors (Table 9).

There is no statistically significant difference in PFS in those with recurrence at more than 1 year in second-line patients by treatment after CDK4/6 inhibitors (Table 9).

There is no statistically significant difference in OS in those with recurrence at more than 1 year in second-line patients by treatment after CDK4/6 inhibitors (Table 9).

There is no statistically significant difference in PFS in those with recurrence at more than 1 year in third-line patients by treatment after CDK4/6 inhibitors (Table 9).

There is no statistically significant difference in OS in those with recurrence at more than 1 year in third-line patients by treatment after CDK4/6 inhibitors (Table 9).

There is a statistically significant difference in OS in those with progression longer than 6 months while receiving CDK4/6 inhibitor therapy, according to the type of subsequent treatment in the first-line setting (*p* = 0.029) (Table 10).

As shown in Figure 3, overall survival (OS) of patients who progressed after CDK4/6 inhibitor treatment is illustrated.

### 3.5. Comparison of mTOR Inhibitor + Endocrine Treatment with Chemotherapy in Visceral Metastatic Patients

When the metastatic site that developed after CDK4/6 inhibitor treatment was visceral disease, no significant difference was found in PFS compared to treatment groups after CDK4/6 inhibitors as shown in Table 11 and Table 12.

Similarly, when the metastatic site that developed after CDK4/6 inhibitor treatment was visceral disease, no significant difference was found in OS compared to treatment groups after CDK4/6 inhibitors as shown in Table 11 and Table 12.

There was no statistically significant difference in PFS after CDK4/6 inhibitor treatment compared to subsequent treatments in patients with visceral metastasis at the time of diagnosis as shown in Table 12.

There was no statistically significant difference in OS after CDK4/6 inhibitor treatment compared to subsequent treatments in patients with visceral metastasis at the time of diagnosis as shown in Table 12.

### 3.6. Survival Differences by Type of Endocrine Treatment Given with mTOR Inhibitors

No significant difference was found in PFS by endocrine treatment with mTOR inhibitors (Table 13).

No significant difference was found in OS by endocrine treatment with mTOR inhibitors (Table 13).

### 3.7. Multivariate Analysis of Factors Affecting Overall Survival

In the univariate Kaplan–Meier analysis of risk factors affecting overall survival, de novo disease, PR status, progression during CDK4/6 inhibitor treatment, recurrence timing, and post-CDK4/6 inhibitor therapy were found to be significant risk factors. When the significant variables were further evaluated using multivariate stepwise backward (LR) Cox regression analysis, only the presence of de novo disease remained an independent risk factor in the final step. Patients with de novo disease had a 2.65-fold higher risk of death compared to those without de novo disease (HR = 2.65).

As shown in Table 14, the multivariate Cox regression analysis identified de novo disease as an independent risk factor for overall survival.

Below is the graphical representation (forest plot) of the multivariate Cox regression analysis. Each red dot represents the hazard ratio (HR), and the black lines indicate the 95% confidence interval (Cl).

As shown in Figure 4 Patients with de novo disease had a 2.65-fold higher risk of death compared to those without de novo disease (HR = 2.65).

The results of the final multivariate Cox regression analysis for overall survival are summarized in (Table 15).

### 3.8. Safety

When side effects were analyzed according to treatment groups, neutropenia was the most common adverse effect observed in the chemotherapy (CT) group and was found in 61.3% (n = 19) of the patients.

The majority of neutropenia cases were grade 1 (n = 13) and grade 2 (n = 6); grade 3 neutropenia was not observed. In the endocrine treatment and mTOR inhibitor (ET + mTORI) group, neutropenia was much less common (11.1%; n = 3) and was grade 1 in only two patients and grade 2 in one.

Stomatitis was observed at a rate of 9.7% (n = 3; one grade 1, two grade 2) in the CT group and significantly higher in the ET + mTORi group (48.1%; n = 13; 2 grade 1, 11 grade 2).

Diarrhea was reported at a rate of 25.8% (n = 8; two grade 1, six grade 2) in the CT group and 18.5% (n = 5; one grade 1, four grade 2) in the ET + mTORi group.

Arthralgia was rarely observed in the CT group (3.2%; n = 1, grade 1), while it was more frequent in the ET + mTORi group (22.2%; n = 6) and distributed in all grades (two grade 1, two grade 2, two grade 3).

No serious toxicity of grade 3 or higher was found for these side effects in the CT group. In general, there were significant differences in the profile and severity of side effects between the treatment groups, with stomatitis and arthralgia being more frequent and severe in the ET + mTORi group and neutropenia being more common in the CT group. During the study, no patients experienced discontinuation or interruption of treatment.

## 4. Discussion

In this study, the clinical and pathologic features and survival outcomes of different treatment regimens (chemotherapy [CT], everolimus + hormonotherapy [HT], and Other) after CDK4/6 inhibitors were analyzed. Our findings showed that there were no significant differences among treatment groups in key demographic and biological parameters such as age, number of metastases, and KI67 levels, suggesting that the groups were comparable at baseline. Similarly, in large studies such as those by Turner et al. (2018) and Hortobagyi et al. (2020), it has been reported that the demographic characteristics of patients receiving CDK4/6 inhibitor treatment had a similar distribution [17,18].

In our study, significant differences were found among groups in terms of recurrent disease rate and PR positivity. A higher proportion of PR positive patients, especially in the Everolimus + HT group, supports a preference for mTOR inhibitor–containing therapies in patients with high hormone sensitivity. This result is consistent with the results of the BOLERO-2 study reported by Baselga et al. (2012), where the combination of Everolimus + Exemestane was found to be effective in hormone-sensitive advanced breast cancer [19].

No significant difference was observed among treatment groups in terms of PFS and OS, in general, in survival analyses. However, in the subgroup of patients with recurrence at ≥1 year and progression after >6 months on CDK4/6 inhibitor, it is noteworthy that OS duration was longer in the Everolimus + HT treatment group than in the CT group. This finding is in line with previous reports showing that mTOR pathway inhibition may provide a survival advantage in progressive disease [20,21]. For example, Rugo et al. (2019) reported that Everolimus-containing regimens may be beneficial in some resistant subgroups [22].

In particular, among patients with progression occurring later than 6 months during CDK4/6 inhibitor therapy, a statistically significant OS difference was observed according to the type of subsequent first-line treatment, favoring the Everolimus + HT group. This suggests that patients with prolonged benefit from CDK4/6 inhibitors, who likely retain endocrine sensitivity, may achieve superior outcomes with mTOR inhibitor–based endocrine combinations rather than chemotherapy. Such findings underscore the potential value of tailoring post-CDK4/6 inhibitor therapy to disease biology and prior response characteristics.

In summary, Everolimus combined with hormonotherapy was shown to be the most suitable option, especially for metastatic patients with late progression following first-line CDK4/6 inhibitor therapy and without known actionable mutations.

In our study, the overall PFS was approximately 19.6 months with CT, 44.7 months with Everolimus + HT, and 25.8 months with hormone therapy alone. While this figure clearly exceeds the 6–8 months generally reported in the literature for this treatment setting, several factors may explain this difference. First, the Everolimus + HT group in our cohort included a relatively high proportion of patients with endocrine-sensitive disease (recurrence ≥12 months after adjuvant endocrine therapy) and high PR positivity, both of which are known to be associated with prolonged benefit from endocrine-based therapies. Second, the rate of visceral crisis was low in this subgroup, which may have contributed to better outcomes compared with less selected real-world cohorts. Taken together, these favorable baseline characteristics likely explain the longer PFS observed and underscore the importance of considering patient selection when interpreting survival results.

Similarly, in multicenter observational analyses in the literature, it has been reported that, among patients treated with Everolimus + HT following CDK4/6 inhibitor therapy, the benefit from treatment was more pronounced in those without visceral metastases and with prolonged endocrine sensitivity [20,23]. In large cohort studies, it has been emphasized that there is no significant difference between CT and Everolimus + ET in terms of PFS and OS, and that the choice of treatment may vary according to patient characteristics [21,24]. However, chemotherapy remains an important treatment option in patients with visceral crisis or rapid progression.

In our study, no statistically significant difference was observed between CT and Everolimus + ET in terms of PFS or OS across all lines of CDK4/6 inhibitor therapy; however, a numerical advantage was noted in favor of the Everolimus + ET group. The fact that no chemotherapy superiority was observed in patient subgroups with visceral organ involvement is meaningful in terms of treatment selection. Thus, chemotherapy can be pushed to the background, and unnecessary exposure can be avoided, considering its side effects.

A retrospective study in the US showed that Everolimus + HT provided a significant PFS and OS advantage in patients with less aggressive clinical profiles compared to chemotherapy [17,25]. However, the biggest shortcoming of this study is that there was no prior use of cdk4/6 inhibitors, and detailed subgroup analyses such as time to recurrence, time to progression, and time to progression during CDK4/6 inhibitor treatment were not performed.

Based on our data analysis, OS and PFS did not differ according to the type of endocrine therapy combined with the mTOR inhibitor.

No randomized clinical trial has directly compared exemestane combinations to Fulvestrant + Everolimus.

Although there are no direct randomized comparisons, retrospective analyses suggest that the combination of everolimus + exemestane may show higher efficacy in terms of PFS than regimens combined with fulvestrant, especially in patient groups similar to ours [19,26].

Although specific comparisons for the combination of Fulvestrant + Everolimus are limited, it provides similar internal data in terms of PFS and OS, making it an important option, especially for post-CDK4/6 inhibitor use.

Our findings seem to be consistent with the literature, in that there is no difference in PFS or OS between faslodex (fulvestrant) and exemestane in combination with Everolimus.

In addition, the multivariate analysis performed in our study demonstrated that only de novo disease emerged as an independent prognostic factor for overall survival. This highlights the importance of baseline disease presentation and suggests that patients with de novo metastatic disease may have poorer outcomes, regardless of subsequent treatment choice.

Limitations of our study include its retrospective design and the small number of patients in some of the subgroup analyses. Further prospective, multicenter studies are needed. However, in general, this multicenter study stands out for its detailed subgroup analyses.

In conclusion, while treatment options after CDK4/6 inhibitor treatment yielded similar results in terms of overall survival and progression, the Everolimus + HT combination may provide a survival advantage in subgroups with certain clinical characteristics. This supports the importance of personalized treatment approaches.

## 5. Conclusions

In patients without mutations relevant for targeted therapy, second-line treatment remains a controversial issue. Therefore, results obtained from real-world data are of particular importance. Our study provided meaningful findings through detailed subgroup analyses. In particular, a significant survival advantage in favor of mTOR inhibitors plus endocrine therapy was observed in patients who were not at an advanced stage at initial diagnosis but relapsed ≥12 months after completion of adjuvant endocrine therapy (endocrine-sensitive), as well as in the subgroup that developed progression at >6 months while receiving a CDK4/6 inhibitor. Similar survival advantages were achieved with chemotherapy in cases with visceral involvement, such as that in the lung and liver, provided that no visceral crisis was present. These findings are clinically relevant to avoid unnecessary exposure to chemotherapy and its potential adverse effects.

According to the univariate analyses, several clinical factors appeared to influence the outcomes; however, in the multivariate model, only de novo disease emerged as an independent prognostic risk factor, underscoring the importance of baseline disease presentation in long-term outcomes.

Moreover, the detailed presentation of subgroup analyses in this study provides clinicians with a clearer perspective for treatment selection. Nevertheless, larger prospective studies are still required to confirm these findings and guide more standardized treatment strategies in this patient population.

## Figures and Tables

**Figure 1 jcm-14-07376-f001:**
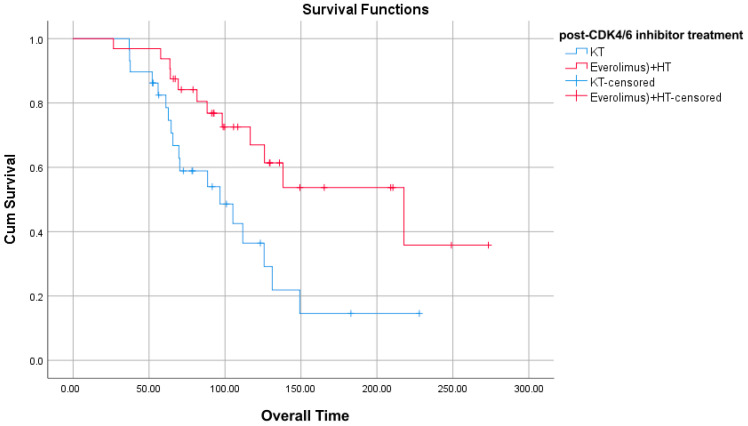
OS graph for those with recurrence at ≥1 year.

**Figure 2 jcm-14-07376-f002:**
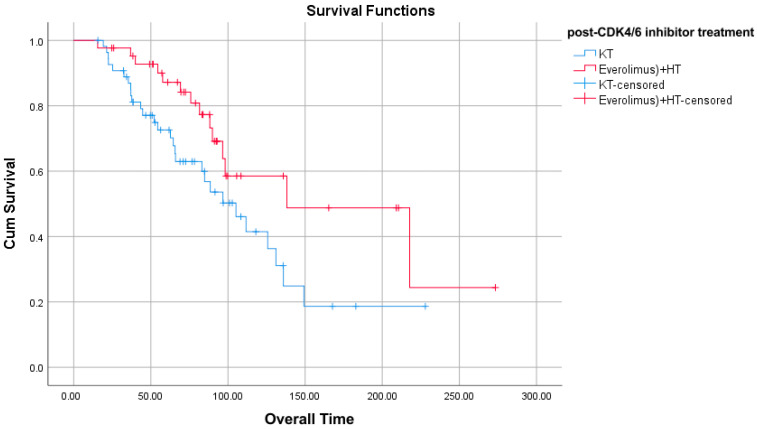
OS in patients with progression after 6 months while receiving CDK 4/6 inhibitors.

**Figure 3 jcm-14-07376-f003:**
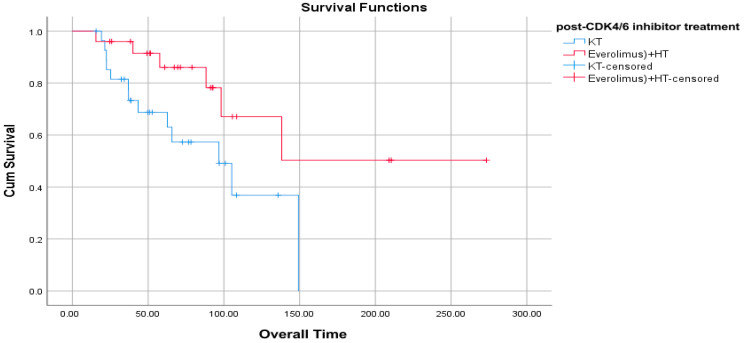
OS of those with progression > 6 months as first-line while receiving CDK4/6 inhibitor treatment.

**Figure 4 jcm-14-07376-f004:**
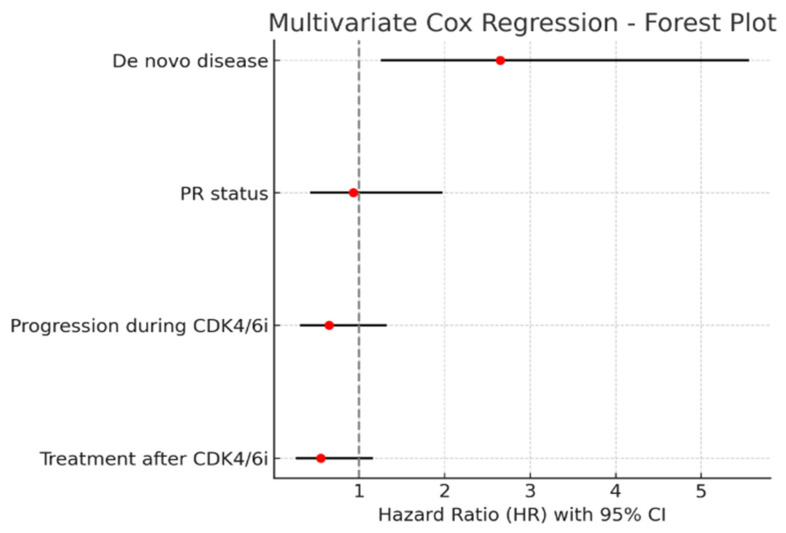
Forest plot showing the hazard ratio (HR) for overall survival according to treatment after CDK4/6 inhibition.

**Table 1 jcm-14-07376-t001:** Comparisons by treatment after CDK4/6 inhibitor treatment.

Characteristic	CT (n = 76)	Mean ± SD/Median (Min–Max)	Everolimus + ET (n = 58)	Mean ± SD/Median (Min–Max)	Other (n = 11)	Mean ± SD/Median (Min–Max)	*p*-Value
Age (years)	76	50.82 ± 12.68	58	47.84 ± 12.02	11	49.99 ± 16.02	0.965
Number of metastatic sites	76	2 (1–6)	58	2 (1–5)	11	1 (1–3)	0.124
Ki-67 (%)	66	20 (5–90)	45	20 (5–70)	8	27.5 (10–50)	0.687

No statistically significant differences were observed across groups.

**Table 2 jcm-14-07376-t002:** Comparisons by treatment after CDK4/6 inhibitor treatment.

Variable	CT n (%)	Everolimus + ET n (%)	Other n (%)	Overall *p*	Pairwise *p*
Menopausal status				0.384	–
Pre	22 (28.9%)	22 (37.9%)	4 (36.4%)		
Peri	6 (7.9%)	1 (1.7%)	1 (9.1%)		
Post	48 (63.2%)	35 (60.3%)	6 (54.5%)		
Pathological diagnosis				0.940	–
IDC	55 (72.4%)	44 (75.9%)	10 (90.9%)		
ILC	11 (14.5%)	9 (15.5%)	1 (9.1%)		
Mixed	4 (5.3%)	3 (5.2%)	0 (0.0%)		
Other	6 (7.9%)	2 (3.4%)	0 (0.0%)		
Recurrent disease				0.027	CT–E: 0.429; CT–O: 0.021; E–O: 0.009
No	34 (44.7%)	22 (37.9%)	9 (81.8%)		
Yes	42 (55.3%)	36 (62.1%)	2 (18.2%)		
Recurrence timing				0.112	–
<1 year	13 (31.0%)	6 (15.8%)	-		
≥1 year	29 (69.0%)	32 (84.2%)	-		

**Table 3 jcm-14-07376-t003:** Stage and T/N distribution by treatment.

Variable	CT n (%)	Everolimus + ET n (%)	Other n (%)	*p*-Value
Stage				0.240
I	3 (3.9%)	1 (1.7%)	0 (0.0%)	
II	18 (23.7%)	22 (37.9%)	1 (9.1%)	
III	11 (14.5%)	10 (17.2%)	1 (9.1%)	
IV	44 (57.9%)	25 (43.1%)	9 (81.8%)	
T category				0.447
T1	11 (14.5%)	11 (19.0%)	4 (36.4%)	
T2	51 (67.1%)	36 (62.1%)	4 (36.4%)	
T3	12 (15.8%)	9 (15.5%)	3 (27.3%)	
T4	2 (2.6%)	2 (3.4%)	0 (0.0%)	
N category				0.913
N0	13 (17.1%)	10 (17.2%)	2 (18.2%)	
N1	43 (56.6%)	36 (62.1%)	8 (72.7%)	
N2	16 (21.1%)	11 (19.0%)	1 (9.1%)	
N3	4 (5.3%)	1 (1.7%)	0 (0.0%)	

**Table 4 jcm-14-07376-t004:** Comparisons by treatment after CDK4/6 inhibitor treatment.

Variable	CT n (%)	Everolimus + HT n (%)	Other n (%)	Overall *p*	Pairwise *p*
Primary operated				0.157	–
No	39 (51.3%)	22 (37.9%)	7 (63.6%)		
Yes	37 (48.7%)	36 (62.1%)	4 (36.4%)		
Adjuvant CT				0.099	–
No	58 (76.3%)	37 (63.8%)	10 (90.9%)		
Yes	18 (23.7%)	21 (36.2%)	1 (9.1%)		
Adjuvant HT				0.085	–
None	45 (59.2%)	26 (44.8%)	9 (81.8%)		
Tamoxifen	18 (23.7%)	20 (34.5%)	0 (0.0%)		
Aromatase inhibitor	13 (17.1%)	12 (20.7%)	2 (18.2%)		
AI type				0.063	–
Letrozole	11 (84.6%)	10 (83.3%)	0 (0.0%)		
Exemestane	0 (0.0%)	0 (0.0%)	1 (50.0%)		
Anastrazole	2 (15.4%)	2 (16.7%)	1 (50.0%)		
De novo disease				0.023	CT–E: 0.058; CT–O: 0.189; E–O: 0.014
No	32 (42.1%)	34 (58.6%)	2 (18.2%)		
Yes	44 (57.9%)	24 (41.4%)	9 (81.8%)		
Treatment received in the metastatic first-line				0.836	–
CT	27 (35.5%)	20 (34.5%)	2 (18.2%)		
CDK4/6 inhibitor	39 (51.3%)	31 (53.4%)	8 (72.7%)		
Hormonotherapy	10 (13.2%)	7 (12.1%)	1 (9.1%)		
Metastatic area at the time of diagnosis				0.211	–
Bone only	12 (15.8%)	13 (22.4%)	5 (45.5%)		
Bone + lymph node	11 (14.5%)	9 (15.5%)	3 (27.3%)		
Internal organs only	14 (18.4%)	10 (17.2%)	0 (0.0%)		
Bone + internal organ	39 (51.3%)	26 (44.8%)	3 (27.3%)		
Progression while receiving CDK4/6 inhibitor				0.551	–
≤6 months	21 (27.6%)	14 (24.6%)	1 (11.1%)		
>6 months	55 (72.4%)	43 (75.4%)	8 (88.9%)		
Endocrine with mTOR					–
Faslodex	-	18 (31.6%)	-		
Exemestane	-	39 (68.4%)	-		
Treatment response after CDK4/6 inhibitor				0.180	–
PR	6 (8.0%)	10 (17.9%)	1 (9.1%)		
SD	13 (17.3%)	13 (23.2%)	4 (36.4%)		
PD	56 (74.7%)	33 (58.9%)	6 (54.5%)		
Post-CDK metastatic area				0.582	–
None	57 (75.0%)	42 (72.4%)	8 (72.7%)		
Bone only	4 (5.3%)	3 (5.2%)	0 (0.0%)		
Bone + lymph node	2 (2.6%)	0 (0.0%)	1 (9.1%)		
Internal organs only	13 (17.1%)	13 (22.4%)	2 (18.2%)		
PR status				0.002	CT–E: 0.001; CT–O: 0.096; E–O: 1.000
Negative	27 (35.5%)	6 (10.3%)	1 (9.1%)		
Positive	49 (64.5%)	52 (89.7%)	10 (90.9%)		
CDK4/6 inhibitor line				0.836	–
First-line	39 (51.3%)	31 (53.4%)	8 (72.7%)		
Second-line	18 (23.7%)	13 (22.4%)	1 (9.1%)		
≥Third-line	19 (25.0%)	14 (24.1%)	2 (18.2%)		

**Table 5 jcm-14-07376-t005:** Comparisons of PFS and OS by treatment after CDK4/6 inhibitor treatment.

Treatment Group	n	PFS (Mean ± SE, 95% CI)	*p*-Value (PFS)	OS (Mean ± SE, 95% CI)	*p*-Value (OS)
CT	76	19.58 ± 2.44 (14.78–24.38)	0.157	111.71 ± 11.28 (89.59–133.83)	0.269
Everolimus + ET	57/58	44.73 ± 5.83 (33.29–56.18)		155.36 ± 16.73 (122.56–188.15)	
Other	11	25.81 ± 5.58 (14.86–36.77)		107.68 ± 16.90 (74.56–140.81)	

Other group: hormonotherapy alone (Fulvestrant or Exemestane). PFS and OS durations are expressed in months. Median survival could not be calculated for several subgroups; therefore, mean survival times with standard errors (SE) and 95% confidence intervals (CI) are reported.

**Table 6 jcm-14-07376-t006:** Comparisons by treatment after CDK4/6 inhibitors by line of use.

Treatment-Line	n	PFS (Mean ± SE, 95% CI)	*p*-Value	OS (Mean ± SE, 95% CI)	*p*-Value
First-line—CT	39	19.70 ± 3.03 (13.76–25.64)	0.403	92.51 ± 10.38 (72.16–112.86)	0.095
First-line—Everolimus + HT	31	26.72 ± 3.70 (19.46–33.98)	-	179.24 ± 25.29 (129.66–228.83)	-
First-line—Other	9	27.43 ± 6.58 (14.52–40.33)	-	63.54 ± 12.80 (38.44–88.64)	-
Second-line—CT	18	14.61 ± 4.23 (6.31–22.92)	0.262	121.87 ± 19.69 (83.26–160.48)	0.495
Second-line—Everolimus + HT	13	23.09 ± 5.78 (11.75–34.43)	-	159.29 ± 30.62 (99.27–219.31)	-
Third-line—CT	19	16.98 ± 4.25 (8.65–25.32)	0.576	107.86 ± 12.49 (83.36–132.36)	0.853
Third-line—Everolimus + HT	14	37.21 ± 11.13 (15.39–59.02)	-	120.60 ± 22.00 (77.46–163.73)	-

Comparisons by treatment after CDK4/6 inhibitors according to treatment line. Data for the “Other” group (hormonotherapy alone: Fulvestrant or Exemestane) for the second and third lines were not included in the analysis due to n < 4.

**Table 7 jcm-14-07376-t007:** PFS and OS by recurrence time.

Subgroup	n	PFS (Mean ± SE, 95% CI)	*p*-Value	OS (Mean ± SE, 95% CI)	*p*-Value
Recurrence < 1 year—CT	13	12.04 ± 3.26 (5.64–18.44)	0.054	92.45 ± 16.50 (60.11–124.79)	0.560
Recurrence < 1 year—Everolimus + HT	6	64.39 ± 16.54 (31.96–96.82)	-	57.67 ± 12.67 (32.82–82.52)	-
Recurrence ≥ 1 year—CT	29	17.19 ± 3.65 (10.03–24.36)	0.115	108.78 ± 13.05 (83.20–134.35)	0.010
Recurrence ≥ 1 year—Everolimus + HT	32	23.61 ± 3.60 (16.55–30.67)	-	180.38 ± 19.28 (142.59–218.17)	-

**Table 8 jcm-14-07376-t008:** PFS and OS by progression timing under CDK4/6 inhibitor treatment.

Subgroup	n	PFS (Mean ± SE, 95% CI)	*p*-Value	OS (Mean ± SE, 95% CI)	*p*-Value
Progression ≤ 6 months—CT	21	12.53 ± 3.12 (6.42–18.65)	0.073	91.54 ± 9.57 (72.77–110.31)	0.691
Progression ≤ 6 months—Everolimus + HT	14	25.14 ± 5.96 (13.45–36.83)	-	135.86 ± 26.15 (84.60–187.13)	-
Progression >6 months—CT	55	21.44 ± 2.80 (15.95–26.93)	0.475	110.27 ± 11.94 (86.87–133.68)	0.039
Progression >6 months—Everolimus + HT	43	44.37 ± 6.81 (31.02–57.72)	-	163.72 ± 20.09 (124.33–203.11)	-

**Table 9 jcm-14-07376-t009:** PFS and OS by line in the recurrence ≥ 1 year subgroup.

Subgroup	n	PFS (Mean ± SE, 95% CI)	*p*-Value	OS (Mean ± SE, 95% CI)	*p*-Value
First-line—CT	15	20.95 ± 4.88 (11.37–30.53)	0.340	99.92 ± 13.12 (74.20–125.63)	0.055
First-line—Everolimus + HT	20	28.24 ± 4.67 (19.09–37.39)	-	193.52 ± 26.40 (141.76–245.28)	-
Second-line—CT	8	15.85 ± 6.76 (2.59–29.11)	0.712	126.88 ± 23.04 (81.71–172.06)	0.500
Second-line—Everolimus + HT	7	17.16 ± 6.42 (4.56–29.76)	-	167.88 ± 33.94 (101.34–234.42)	-
Third-line—CT	6	4.56 ± 1.63 (1.36–7.77)	0.165	69.43 ± 5.95 (57.76–81.10)	0.097
Third-line—Everolimus + HT	5	13.41 ± 6.95 (0.00–27.04)	-	168.68 ± 34.65 (100.75–236.60)	-

**Table 10 jcm-14-07376-t010:** PFS and OS by line in the progression > 6 months subgroup.

Subgroup	n	PFS (Mean ± SE, 95% CI)	*p*-Value	OS (Mean ± SE, 95% CI)	*p*-Value
First-line—CT	28	23.30 ± 3.66 (16.12–30.47)	0.688	92.26 ± 11.83 (69.07–115.45)	0.029
First-line—Everolimus + HT	25	27.17 ± 4.14 (19.06–35.28)	-	184.05 ± 29.21 (126.79–241.31)	-
Second-line—CT	15	16.24 ± 5.05 (6.33–26.15)	0.649	115.39 ± 19.77 (76.62–154.15)	0.277
Second-line—Everolimus + HT	10	16.72 ± 4.99 (6.93–26.52)	-	133.99 ± 14.98 (104.62–163.36)	-
Third-line—CT	12	12.58 ± 2.50 (7.67–17.49)	0.864	112.04 ± 14.91 (82.80–141.28)	0.966
Third-line—Everolimus + HT	8	38.65 ± 14.88 (9.47–67.82)	-	129.06 ± 31.23 (67.83–190.29)	-

**Table 11 jcm-14-07376-t011:** PFS and OS comparisons according to treatment after CDK4/6 inhibitors in patients who developed visceral metastasis post-CDK4/6 inhibitor treatment.

Treatment	n	PFS (Mean ± SE, 95% CI)	*p*-Value (PFS)	OS (Mean ± SE, 95% CI)	*p*-Value (OS)
CT	13	6.55 ± 0.84 (4.90–8.21)	0.989	63.67 ± 10.81 (42.48–84.87)	0.254
Everolimus + HT	13	6.02 ± 1.14 (3.79–8.26)		89.12 ± 12.98 (63.67–114.56)	

**Table 12 jcm-14-07376-t012:** PFS and OS comparisons according to treatment after CDK4/6 inhibitors in patients with visceral metastasis at the time of diagnosis.

Treatment	n	PFS (Mean ± SE, 95% CI)	*p*-Value (PFS)	OS (Mean ± SE, 95% CI)	*p*-Value (OS)
CT	14	21.82 ± 5.42 (11.19–32.46)	0.808	97.58 ± 10.46 (77.07–118.09)	0.131
Everolimus + HT	10	22.96 ± 5.71 (11.76–34.15)		192.30 ± 34.90 (123.88–260.72)	

**Table 13 jcm-14-07376-t013:** Comparison of PFS and OS by endocrine therapy combined with mTOR inhibitors.

Endocrine Agent	n	PFS (Mean ± SE, 95% CI)	*p*-Value (PFS)	OS (Mean ± SE, 95% CI)	*p*-Value (OS)
Faslodex	18	25.81 ± 4.73 (16.53–35.10)	0.819	103.75 ± 16.50 (71.40–136.10)	0.102
Exemestane	38/39	44.42 ± 7.08 (30.53–58.31)		177.17 ± 20.27 (137.52–216.81)	

**Table 14 jcm-14-07376-t014:** Multivariate Cox Regression Model for Overall Survival (Final Model).

Variable	*p*-Value	HR	95% CI for HR
De novo disease	0.009	2.65	1.27–5.55
PR status	0.855	0.93	0.45–1.96
Progression timing under CDK4/6i (≤6 months vs. >6 months)	0.237	0.66	0.33–1.32
Recurrence timing (<1 year vs. ≥1 year)	0.848	1.12	0.34–3.70
Treatment after CDK4/6inhibitör	0.112	0.55	0.27–1.15

Abbreviations: HR, hazard ratio; CI, confidence interval; CDK4/6inhibitor, cyclin-dependent kinase 4/6 inhibitor; PR, progesterone receptor.

**Table 15 jcm-14-07376-t015:** Final Multivariate Cox Regression Model for Overall Survival.

Variable	*p*-Value	HR	95% CI for HR
De novo disease (No vs. Yes)	0.009	2.65	1.27–5.55

Abbreviations: HR, hazard ratio; CI, confidence interval.

## Data Availability

All data generated or analyzed during this study are included in this article. Additional anonymized data may be made available from the corresponding author on reasonable request.

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
