# Peer review of "Efficacy of Subsequent Therapy in Patients with Hormone-Positive Advanced Breast Cancer with Disease Progression After CDK4/6 Inhibitor Therapy: Multicenter Real-Life Data"

_jcm, 2025, doi:10.3390/jcm14207376_

Round 1
Reviewer 1 Report
Comments and Suggestions for Authors
This is a multi-center, real-world study of post-CDK4/6 options in HR+/HER2− MBC. The study addresses a pertinent question, and the data collected provides valuable insights into post-CDK4/6 treatment options. The current approaches to analyzing the data—time-to-event averages, unadjusted survival, and poor subgroup treatment—hinder the reliability of effect estimates. The following adjustments are necessary:
- Page 4, lines 160–162 (starts “Survival analyses were performed by Kaplan-Meier. The descriptive statistics of survival times are given as mean ± standard error.”):
Replace “mean ± standard error” with median with 95% CI from Kaplan–Meier.
- Page 4, lines 151–163: add a sentence defining time-zero and censoring rules for PFS and OS.
Author Response
In some data, the survival median time and its standard error, and therefore the confidence interval, could not be calculated.
Therefore, the confidence interval were calculated according to the survival mean and added to the tables.
The median survival time is calculated as the smallest survival time for which the survivor function is less than or equal to 0.5. Some data sets may not get this far, in which case their median survival time is not calculated.
Reviewer 2 Report
Comments and Suggestions for Authors
This article references and summarizes real-world evidence and data on breast cancer progression, specifically in patients treated with CDK 4/6 inhibitors. The introduction and the body of the text need substantial improvement due to the incorrect use of abbreviations and poor bibliographic referencing.
While the article mentions results from clinical trials with patients who do not have mutations, it doesn't provide a clear guideline for those who don't. For instance, it doesn't adequately analyze whether the most appropriate treatment would be hormone therapy as a monotherapy, hormone therapy combined with an mTOR inhibitor, or chemotherapy, as the text itself suggests. A deeper analysis could have provided valuable conclusions, as the series of over 100 patients is quite interesting. However, the insufficient data analysis makes the results insignificant.
Furthermore, the article should have differentiated the treated patients based on their sensitivity to hormone therapy:
-
Hormone-sensitive patients (those who are in poorer condition or have not received previous treatments).
-
Resistant patients (disease progression within two years after adjuvant treatment).
Analyzing this data in a differentiated manner would be crucial for obtaining more robust results.
Regarding the results, the presentation is confusing, and the data is mixed. The study does not classify the disease in a standardized way. The expression "internal organs" is not a common clinical term. A more precise classification should have been used, such as non-visceral disease or visceral disease. The tables are difficult to read and unclear, and Kaplan-Meier curves would be much more illustrative for demonstrating the experience of the different patient groups.
Comments on the Quality of English Language
In conclusion, this article requires a thorough revision and a clear improvement in its methodology, data analysis, and presentation before it can be accepted for publication.
Author Response
Kaplan–Meier curves are presented only for comparisons with statistically significant differences in OS or PFS, as inclusion of non-significant results would have led to unnecessary complexity and a less clear presentation.
“Recurrence >1 year refers to patients who relapsed ≥12 months after the completion of adjuvant endocrine therapy and are therefore considered endocrine-sensitive. Recurrence <1 year refers to patients who relapsed within the first 12 months after the completion of adjuvant endocrine therapy and represents endocrine-resistant patients” This specification has been incorporated into the Methods, subsection 2.3.
The term “internal organs” was revised to the correct expression “visceral disease.
We thank the reviewer for this valuable comment. In the revised version, the Conclusion section has been substantially strengthened to better reflect the depth and clinical relevance of our findings. We clarified that endocrine-sensitive patients, defined as those who relapsed ≥12 months after completion of adjuvant endocrine therapy, and patients who progressed >6 months on CDK4/6 inhibitors derived a survival advantage from mTOR inhibitor plus endocrine therapy. We also highlighted that in cases with visceral involvement such as lung or liver, chemotherapy provided similar outcomes in the absence of visceral crisis, underscoring its role in selected patients. Furthermore, we incorporated the results of our survival analyses, noting that while several variables appeared significant in univariate analyses, only de novo disease remained as an independent prognostic factor in the multivariate model. These additions improve the clarity of the conclusions, provide a stronger interpretation of the data, and underline the importance of our real-world findings for clinical practice.
The findings from the multivariate analysis have been suitably integrated into the Discussion section
We respectfully disagree with the reviewer’s comment that no clear statement was provided for patients without mutations. In fact, several explicit sentences in the Introduction, Discussion, and Conclusion highlight that in patients without actionable mutations, second-line treatment remains controversial, that no category 1 recommendation exists in guidelines, and that our real-world data address this gap. For example: ‘In patients without mutations relevant for targeted therapy, second-line treatment remains a controversial issue’ (Conclusion). These statements, which we have now emphasized more clearly in the revised manuscript, directly address the reviewer’s concern.
Reviewer 3 Report
Comments and Suggestions for Authors
Dear Authors,
Thank you for the opportunity to review your manuscript, titled “Efficacy of Subsequent Therapy in Patients with Hormone-Positive Advanced Breast Cancer with Disease Progression Under CDK4/6 Inhibitor Therapy: Multicenter Real Life Data”. This is a highly relevant and clinically important topic, and you are to be commended for undertaking a multicenter study to address the complex question of treatment sequencing following CDK4/6 inhibition. Your efforts to perform detailed subgroup analyses are particularly valuable.
Please find below my comments and suggestions, which I hope you will find constructive for strengthening your manuscript.
1. Introduction:
The introduction provides a good overview of the treatment landscape and correctly identifies the knowledge gap this study aims to fill. However, it could be more sharply focused by explicitly stating the primary objective and hypotheses of this specific study in the closing paragraph, rather than a general aim to "determine the appropriate treatment."
2. Research Design:
The retrospective, multicenter design is appropriate for generating real-world evidence on this question. The primary limitation inherent to this design is the potential for selection bias and confounding factors, which is acknowledged in the discussion. The definition of the patient population is clear, but a critical methodological detail is missing (see point 4).
3. Methods:
The methodology requires crucial clarifications to ensure the validity and interpretability of the findings.
-
The composition of the “Other treatments” group is undefined beyond “monotherapy endocrine therapy or different agents.” A precise breakdown of the specific agents and the number of patients receiving each is essential. The heterogeneity of this group likely confounds its comparisons with the well-defined CT and Everolimus+HT arms.
-
A paramount issue is the exclusion of patients with actionable mutations (e.g., PIK3CA, *BRCA1/2*). The conclusion states the study is for patients with "no mutations," but the methods section must explicitly state the policy regarding molecular testing for all patients. Were patients with these mutations excluded? If not, their inclusion would severely confound the results, as they would be candidates for superior targeted therapies.
4. Results:
The results are extensive but would benefit from greater clarity and validation.
-
The reported median PFS of 44.7 months for the Everolimus+HT group is extraordinarily high, vastly exceeding the 6-8 months typically reported in the literature for this setting. This outlier result requires thorough validation of the data and statistical calculations, and a potential explanation should be offered in the discussion.
-
The presentation of the toxicity data could be enhanced for clarity. The denominators switch from the full cohort (n=76, 58, 11) to percentages that suggest a much smaller safety population (e.g., n=19 for CT neutropenia). Safety data must be presented clearly for the entire cohort, with all-grade and grade ≥3 adverse events listed for each group.
5. Conclusions:
The conclusions are generally supported by the results but should be tempered by the study's limitations. The promising signal in the subgroups is hypothesis-generating and should be framed as such, emphasizing the need for validation in prospective studies. The call for personalized medicine is well-taken.
6. Figures and Tables:
The tables are comprehensive but complex. Some streamlining and improved labeling (e.g., explicitly stating n for each subgroup in survival tables) would enhance clarity. The captions for Tables 2E are duplicated and need correction.
7. English Language:
The English is generally understandable but requires professional editing for grammar and syntax to improve clarity and flow.
-
Example 1: “The rate of recurrent disease was lower in patients than with CT and (Everolimus)+HT.” This sentence is unclear and should be rephrased for meaning.
-
Example 2: “But in general, it was a multicenter study that stood out with detailed subgroup analyses.” Starting a sentence with "But" is informal; "Nevertheless or That said" are more preferable alternatives in scientific writing.
8. Originality/Novelty:
The study addresses a common and pressing clinical dilemma. While the comparison itself is not novel, the detailed subgroup analyses based on recurrence interval and duration of benefit from CDK4/6i provide valuable real-world insights that contribute to the literature.
9. Quality of Presentation:
The manuscript is logically structured. As noted, improvements can be made in the clarity of data presentation in tables and the flow of the language through careful editing.
10. Scientific Soundness:
The statistical methods applied are standard for this study type. However, to strengthen the scientific soundness and account for the identified baseline imbalances, I strongly recommend performing a multivariate Cox regression analysis for the key subgroup findings. Alternatively, propensity score matching could be used to create more balanced cohorts for comparison. This would add robustness to the conclusion that the treatment effect is independent of these confounders.
In summary, your study highlights an important clinical strategy. By addressing these points, particularly the definition of the cohort regarding mutations and the "Other" group, and by strengthening the statistical analysis, you can significantly enhance the impact and credibility of your valuable work.
Thank you again for contributing to this important field.
Sincerely,
The Reviewer
Comments on the Quality of English LanguageThe manuscript requires comprehensive English language editing to enhance clarity, grammatical accuracy, and professional tone. While the scientific content is discernible, numerous language issues presently hinder the reading flow and precision of communication. Key areas for improvement include:
- Article Usage: Missing definite and indefinite articles are common throughout the text (e.g., ...primary recommendation for advanced HR-positive..., should be ...for the management of advanced HR-positive...).
- Prepositions: Incorrect preposition use is frequent (e.g., progressed under adjuvant treatment" could be better "progressed during adjuvant treatment).
- Tense Consistency: There is some inconsistency in verb tenses between and within sentences.
- Sentence Structure: Many sentences are overly long and complex, combining multiple ideas. Breaking these into shorter, more direct sentences would greatly improve readability.
Recommendation: I strongly recommend that the authors seek the assistance of a professional scientific editing service or a native English speaker with expertise in medical writing. This will ensure the manuscript is polished and presents its valuable findings in the clearest and most impactful way possible.
Author Response
1. We thank the reviewer for this valuable comment regarding the clarity of the Introduction. As suggested, we have revised the closing part of the Introduction to explicitly state the primary objective and hypothesis of our study. In the revised version, instead of a general statement about “determining the appropriate treatment,” we now clearly specify that the aim of our study was to evaluate the efficacy of different subsequent treatment regimens (chemotherapy, everolimus + endocrine therapy, and other endocrine-based approaches) in patients with HR+/HER2− metastatic breast cancer who progressed on CDK4/6 inhibitors and who did not harbor actionable mutations. In addition, we have clarified our hypothesis that endocrine-based therapies, particularly everolimus-containing combinations, may provide survival outcomes comparable to or better than chemotherapy in selected subgroups, such as endocrine-sensitive patients or those without visceral crisis. We believe that this revision sharpens the focus of the Introduction and aligns better with the reviewer’s suggestion.
3.We thank the reviewer for this valuable comment. In the revised manuscript, we have clarified this point in the Methods, Section 2.1 Study Design and Patient Disposition. Specifically, we added the following sentence: “Patients with actionable mutations (e.g., PIK3CA, BRCA1/2) were excluded, as they would be candidates for targeted therapies not comparable to the treatment arms investigated in this study.” This addition ensures that the study cohort is clearly defined and avoids potential confounding by targeted therapy–eligible patients.
4.We would like to thank the reviewer for this important comment. We re-checked the reported median PFS value of 44.7 months in the Everolimus + HT group and confirmed that our data are accurate. We are aware that this figure is much higher than the 6–8 months generally reported in the literature (Baselga et al., 2012; Rugo et al., 2019). We believe that this difference is due to the more favorable clinical characteristics of the patient population in our study, such as a higher proportion of endocrine-sensitive patients and the absence of visceral crisis. A statement highlighting this unusual result, comparing it with previous publications, and emphasizing the need for cautious interpretation has been added to the Discussion section.
5.We agree and will temper the conclusions by highlighting the study’s limitations and framing subgroup findings as hypothesis-generating, requiring confirmation in prospective studies.
6. A detailed and rigorous statistical analysis was performed in the study. The large number of subgroup analyses may have given the impression of complexity; They do not describe the same analysis. One table presents outcomes according to the time to progression while patients were still receiving CDK4/6 inhibitors, whereas the other focuses on recurrence time after completion of adjuvant endocrine therapy. The earlier tables reflect the general population, whileThe two “Table 2E” sets present more detailed subgroup analyses according to the line of CDK4/6 inhibitor use
7.English Language: We will revise the manuscript with professional language editing to improve clarity, grammar, and flow. The unclear sentences will be rephrased, and informal wording replaced with more appropriate academic alternatives.
8.Originality/Novelty: We are pleased the reviewer recognizes the value of our work. Although the comparison itself is not novel, the detailed subgroup analyses provide meaningful real-world insights.
9.Quality of Presentation: We will refine table presentation and improve language flow as recommended.
10.As recommended by the reviewer, we conducted a multivariate Cox regression analysis to adjust for potential confounding factors. The results of this analysis have been added to the Results section, and the Discussion was updated accordingly to highlight that only de novo disease emerged as an independent prognostic factor."
Reviewer 4 Report
Comments and Suggestions for Authors
It would be interesting to see further exploration of molecular or genomic markers that could help predict benefit from therapy in current study. Overall, ythis study contributes valuable evidence to a new evolving treatment landscape.
Author Response
With the practical tests we routinely use, we were able to rule out PIK3CA and BRCA. However, it is clear that further advanced research is needed. We sincerely thank you for your valuable comment.
Round 2
Reviewer 2 Report
Comments and Suggestions for Authors
Introduction and Terminology
- Abbreviations: The introduction continues to use inconsistent abbreviations, with a mix of capitalized and lowercase letters. These inconsistencies, which were present in the initial draft, have not been corrected and compromise the document's uniformity.
- CDK Inhibitors: The text discusses CDK inhibitors without adequate specificity. It must clarify whether it refers to CDK 4/6 inhibitors or CDK 2 inhibitors, given the existence of inhibitors beyond the CDK 4/6 class.
Materials and Methods
- Patient Exclusion Criteria: The clarification provided in the Materials and Methods section is appropriate. It clearly states that patients receiving targeted therapies are excluded, and only those without such therapies are included. This improves the homogeneity of the patient cohort.
Results and Presentation
- Tables: The tables, which appear to be unedited outputs from the statistical software, are difficult to interpret and are not suitable for a scientific publication. They require substantial editing and formatting for clarity and professional presentation.
- Hormone-sensitivity Clarification: The explanatory note regarding hormone mono-sensitivity remains confusing and does not clearly specify how patient groups were analyzed based on their sensitivity. Furthermore, the statement that patients who recur within 12 months are included is ambiguous. It implies the inclusion of patients who experience progression during adjuvant treatment, which is not explicitly stated in the main text and needs clarification.
- Abstract/Results Discrepancy: The Abstract states that recurrence rates are higher in patients who receive endocrine therapy and chemotherapy (or a similar combination, assuming "eurolines" is a typo for "endocrine"), which directly contradicts the findings presented in the Results section. This discrepancy must be rectified.
A thorough review of the English composition is necessary. Certain segments display incorrect grammatical construction—specifically, the omission of necessary verbs—and generally phrasing needs improvement. This detracts from the clarity and professionalism required for a scientific publication.
Author Response
1. Abbreviations
We appreciate the reviewer’s observation. All abbreviations throughout the manuscript have been carefully reviewed and standardized for consistency. Terms such as “HR+/HER2−,” “CDK4/6 inhibitor (CDK4/6i),” “ET (endocrine therapy),” and “OS/PFS” are now consistently capitalized and defined upon first use in the introduction. This revision ensures uniformity across the entire document.
2. CDK Inhibitors
We thank the reviewer for this important point. All references to “CDK inhibitors” in the manuscript have been revised to specifically refer to “CDK4/6 inhibitors”, as this study exclusively investigates patients treated with CDK4/6 inhibitors (palbociclib, ribociclib, or abemaciclib). No other CDK inhibitor classes were included. The terminology has been corrected in both the Introduction and Methods sections to avoid ambiguity.
3. Patient Exclusion Criteria
We thank the reviewer for acknowledging this clarification. No further changes were necessary.
4. Tables
We fully agree with the reviewer. All tables have been extensively reformatted for clarity and readability. Unnecessary statistical outputs (e.g., SE, Wald, df) have been removed. The revised tables now include only essential parameters such as p-values, HRs, and 95% CIs, with standardized titles and footnotes. Survival tables have also been harmonized to present results as mean ± SE (95% CI), as median survival could not be reached in several subgroups.
5. Hormone-sensitivity Clarification
We appreciate the reviewer’s valuable feedback. To address this, we have revised Section 2.3 (Treatment Arms and Subgroup Classifications) to include a clear definition:
“Hormone sensitivity was defined according to the time to recurrence following the completion of adjuvant endocrine therapy.
– Patients who relapsed ≥12 months after completing adjuvant endocrine therapy were considered endocrine-sensitive.
– Patients who relapsed within <12 months after completing adjuvant endocrine therapy were considered endocrine-resistant.
Patients who experienced disease progression during adjuvant endocrine therapy were not included in this study, as this subgroup was not analyzed separately.”
This revision eliminates any ambiguity regarding inclusion criteria and clarifies that progression during adjuvant therapy was not part of the analysis.
We have removed the row labeled “progression under adjuvant therapy” from the relevant table, as you correctly pointed out that these cases do not represent true progression or residual disease. Instead, these patients belonged to the recurrence group that relapsed within 12 months after completing adjuvant endocrine therapy.
6. Abstract/Results Discrepancy
We thank the reviewer for pointing out this inconsistency. The word “eurolines” was a typographical error in the previous version. The Abstract has been revised to correctly state:
“The rate of recurrent disease was significantly higher in the CT and Everolimus+HT groups compared to the ‘Other’ group (p = 0.027).”
This now aligns fully with the data presented in the Results section. All discrepancies between the Abstract and Results have been corrected.
7. English Language
We appreciate the reviewer’s advice. The entire manuscript has been thoroughly revised through the MDPI English editing service by a native English-speaking editor with expertise in medical writing to ensure grammatical accuracy, clarity, and a professional tone.
Reviewer 3 Report
Comments and Suggestions for Authors
Dear Authors,
Thank you for your detailed and thoughtful responses to my initial comments. I have carefully reviewed your revisions and clarifications, and I am very pleased with the steps you have taken to improve the manuscript.
You have addressed all my points comprehensively and effectively. Specifically, I would like to highlight the following significant improvements:
-
Sharpened Introduction: The revised closing paragraph, which now explicitly states the primary objective and hypothesis, provides a much clearer focus for the reader.
-
Crucial Methodological Clarification: The explicit statement in the methods regarding the exclusion of patients with actionable mutations is a critical addition that strengthens the validity of your cohort definition and subsequent comparisons.
-
Transparency on Results: Your commitment to acknowledging and discussing the unusually high PFS in the Everolimus+HT group, with a plausible explanation based on patient characteristics, demonstrates excellent scientific rigor.
-
Enhanced Statistical Robustness: The inclusion of a multivariate Cox regression analysis is a major strength. It effectively addresses potential confounding and adds significant weight to your findings, particularly the identification of de novo disease as an independent prognostic factor.
-
Language Polishing: The improved clarity in your responses is evident, and I am confident that the final manuscript will benefit from the professional editing you have undertaken.
The manuscript is now substantially stronger. The revisions have enhanced its clarity, scientific soundness, and overall impact. I believe it represents a valuable contribution to the real-world evidence on this important clinical dilemma.
I congratulate you on your excellent work and have no further main concerns. I wish you the best with the final steps of the process.
Sincerely,
The Reviewer
Author Response
We sincerely thank the reviewer for their thorough re-evaluation and highly encouraging feedback. We are delighted to learn that the revisions have successfully addressed all concerns and improved the clarity, scientific rigor, and overall quality of the manuscript.
We truly appreciate the reviewer’s recognition of our efforts to refine the introduction, strengthen methodological transparency, enhance statistical robustness, and ensure professional language editing.
It is a great honor to receive such positive and constructive remarks. We are grateful for the reviewer’s time, expertise, and valuable guidance throughout the revision process, which have significantly contributed to the final version of our study.